# Controlling AMR in the Pig Industry: Is It Enough to Restrict Heavy Metals?

**DOI:** 10.3390/ijerph191811265

**Published:** 2022-09-07

**Authors:** Na Li, Hongna Li, Changxiong Zhu, Chong Liu, Guofeng Su, Jianguo Chen

**Affiliations:** 1Department of Engineering Physics, Tsinghua University, Beijing 100084, China; 2Institute for Public Safety Research, Tsinghua University, Beijing 100084, China; 3Institute of Environment and Sustainable Development in Agriculture, Chinese Academy of Agricultural Sciences, Beijing 100081, China

**Keywords:** heavy metal, metagenomic sequencing, soil, pig manure

## Abstract

Heavy metals have the potential to influence the transmission of antimicrobial resistance (AMR). However, the effect on AMR caused by heavy metals has not been clearly revealed. In this study, we used a microcosm experiment and metagenomics to examine whether common levels of Cu and Zn in pig manure influence AMR transmission in manured soil. We found that the abundance of 204 ARGs significantly increased after manure application, even though the manure did not contain antibiotic residuals. However, the combined addition of low Cu and Zn (500 and 1000 mg/kg, respectively) only caused 14 ARGs to significantly increase, and high Cu and Zn (1000 and 3000 mg/kg, respectively) caused 27 ARGs to significantly increase. The disparity of these numbers suggested that factors within the manure were the primary driving reasons for AMR transmission, rather than metal amendments. A similar trend was found for biocide and metal resistance genes (BMRGs) and mobile genetic elements (MGEs). This study offers deeper insights into AMR transmission in relation to the effects of manure application and heavy metals at commonly reported levels. Our findings recommend that more comprehensive measures in controlling AMR in the pig industry are needed apart from restricting heavy metal additions.

## 1. Introduction

Heavy metals, such as Cu and Zn, have been extensively used for the past decades for pigs to improve growth and prevent diseases. These metals promote growth via their antimicrobial nature on the gut microbiota that reduce fermentation loss of nutrients and suppress gut bacteria, including pathogens [1]. In China, heavy metals and antibiotics were used simultaneously, while in Europe, antibiotic restrictions have led to heavy metal addition to feeds as alternatives to antibiotics [2]. Among all of the trace elements, Cu and Zn were found to be the most-abundant heavy metals in pig, cattle, and poultry manure [3]. Cu additions to pig feeds range from 75 to 250 mg/kg according to the age of pigs with low digestibility [4]. Since these metals would be almost entirely excreted, the heavy metal content in animal manure is largely a reflection of their addition to feeds [3,5].

Excessive Cu in the gut exerts its antimicrobial action by damaging the structure of bacterial proteins or DNA and the generation of reactive oxygen species (ROS) causing oxidative stress [4,6]. Metals can also directly and indirectly induce antibiotic resistance genes (ARGs) carried by bacteria in various ecosystems [7]. For example, resistance to multiple antibiotics in CuSO_4_ contaminated soil was significantly enhanced compared with uncontaminated control soil [8]. Higher ARG transfer frequencies were also proved under heavy metal stress in sludge bacterial communities [9]. Moreover, heavy metals do not degrade in the environment, and this recalcitrance can impose a long-standing ARG co-selection pressure [10].

The mechanisms of the co-selection of antibiotic and metal resistance include co-resistance (different resistance determinants present on the same genetic element), cross-resistance (the same genetic determinant responsible for resistance to antibiotics and metals, such as efflux pump genes), and indirect but shared regulatory responses to metal and antibiotic exposure, such as biofilm induction [11]. Although these mechanisms were described decades ago, only in recent years has this hazard been realized, as ARG transmission and heavy metal discharge from pig farms were linked in European countries [12]. The European Union now restricts Cu and Zn addition to pig feeds [12,13], although it is still unclear whether and how these metals influence antimicrobial resistance (AMR) in manured soils. Moreover, previous studies focusing on heavy metal stress mainly used a single element to demonstrate the influence of the element on ARGs, and few have analyzed the effects of combined elements on the transmission of ARGs.

In this study, we designed a microcosm experiment to explore the influence of a combination of Cu and Zn in pig manure on the transmission of ARGs, biocide and metal resistance genes (BMRGs), and mobile genetic elements (MGEs) in the soil following manure application. Cu and Zn were chosen because the two trace elements are the most-abundant heavy metals in pig manure. We analyzed bacterial α diversity and community structure to determine the impact of heavy-metal-contaminated pig manure on the soil bacterial community. This study offers basic knowledge on the influence of the two mostly used heavy metals in pig farming on the transmission of AMR from manure application.

## 2. Materials and Methods

### 2.1. Experimental Design and Sampling

Topsoil (0–20 cm in depth) was collected from a park in Haidian District, Beijing, China, in April 2021. The soil had not been received any application of manure to the best of our knowledge. The soil possessed the following properties: pH 8.33, electrical conductivity 175.5 ± 6.5 μS/cm, total nitrogen 1.02 ± 0.02 g/kg, total phosphorus 0.83 ± 0.03 g/kg, and soil organic carbon 1.56 ± 0.02%. The background heavy metal concentrations in the original soil and pig manure were presented in Table 1. The soil was sieved through a 2 mm mesh to remove plant roots and large particles. Air-dried pig manure was collected from a remote village (26° N, 102° E) on a mountain in Kunming, Yunnan Province, China, where pigs were not intensively raised and where no growth promoters were added to the feeds. Therefore, this manure represents an ideal environmentally friendly organic fertilizer.

A recent review reported that the average concentrations of Cu and Zn in typical pig manure in China were about 500 and 1000 mg/kg, respectively, while those in Japan were approximately 1000 and 3000 mg/kg, respectively [14]. Therefore, these concentration levels could be taken as common concentrations in different regions. In practice, pig manure is usually applied to soil at ~4% *w*/*w* as a fertilizer [15]. Based on these data, four treatments were set in this study as follows: (1) soil without manure or chemical additions (Control) and soils containing 4% *w*/*w* pig manure with the following additions (mg/kg) (2) no added metals (NCuZn), (3) Cu 500 and Zn 1000 (LCuZn), and (4) Cu 1000 and Zn 3000 (HCuZn). Soils (300 g) were placed in sterilized glass jars, and distilled water was added to a 60% water-holding capacity. The jars were sealed with Parafilm and punctured with six holes using a needle to avoid an anaerobic environment. The jars were incubated in an MLR-350HT climatic chamber (Sanyo, Osaka, Japan) at 25 °C in the dark. Treatments were conducted in triplicate. Soils were sampled after 60 days. One proportion of each sample was air-dried for physiochemical testing, while the other proportion was stored at −80 °C before DNA extraction.

### 2.2. Measurements of Heavy Metals in Original Manure and Soil

The metal content of the experimental soils was determined by 12 mL aqua regia (HCl: HNO_3_, *v*:*v* = 3:1) and 10 drops of HClO_4_ digestion at 155 °C until no visible smoke came out and until the samples became pale white (~16 h). Available Cu and Zn were assessed by the extraction of soil samples with 5 mM diethylenetriaminepentaacetic acid (DTPA), 10 mM CaCl_2_, and 100 mM triethanolamine (TEA, pH 7.3) at a soil:liquid ratio of 1:2 *w*/*v*. Cu and Zn concentrations were determined using an Optima 5300 DV inductively coupled plasma optical emission spectrometer (Perkin Elmer, Northampton, MA, USA) as previously described [16].

### 2.3. DNA Extraction from Soils

DNA was extracted from soil samples using the Mag Bind Soil DNA Kit (Omega Bio-Tek, Norcross, GA, USA) in triplicate for each treatment (12 samples in total). DNA quality was assessed using a fluorescence spectrophotometer and a Quant-iT PicoGreen dsDNA Assay Kit P7589 (Invitrogen, Carlsbad, CA, USA) and 1% agarose gel electrophoresis. DNA was stored at −20 °C prior to use.

### 2.4. Metagenomic Sequencing and Bioinformatic Analysis

A whole-genome shotgun approach was used to generate sequencing libraries with average insert sizes of 400 bp using an Illumina TruSeq Nano DNA LT Library Preparation Kit. Then paired-end sequencing (2 × 150 bp) was performed on the Novaseq 6000 system (Illumina, San Diego, CA, USA) at Personal Biotechnology (Shanghai, China). Low-quality sequences and Illumina adapters were filtered out using Cutadapt (v 1.2.1). Sequences were filtered based on size (≥50 bp), and quality scores (≥Q20) and >12 Gbp high-quality data from each soil sample were generated for downstream analysis. The raw sequencing data have been deposited in the NCBI Sequence Read Archive (SRA) under the BioProject PRJNA861428.

Quality-filtered reads were queried to the NCBI RefSeq database using Kraken2 for bacterial taxonomic classifications and sequences [17]. Sequences assigned to Viridiplantae or Metazoa were eliminated. For ARG annotation, the generated contigs (≥200 bp) were compared against the CARD database (https://card.mcmaster.ca/, accessed on 1 December 2019). BMRG compositions were evaluated using the Bacmet2 database (http://bacmet.biomedicine.gu.se, accessed on 1 December 2019). MGE profiles were constructed to identify horizontal gene transfer (HGT) potential using the plasmid sequences downloaded from NCBI RefSeq (https://ftp.ncbi.nlm.nih.gov/genomes/refseq/plasmid/, accessed on 10 December 2019), ISfinder (https://www-is.biotoul.fr/, accessed on 10 December 2019), INTEGRALL (http://integrall.bio.ua.pt, accessed on 1 December 2019), and ICEberg (https://db-mml.sjtu.edu.cn/ICEberg, accessed on 10 December 2019) databases to identify plasmids, insertion sequences (ISs), integrons, and integrative and conjugative elements (ICEs), respectively. While using the above databases, the top one blast result with E value < 0.001 was retained for further analysis.

### 2.5. Statistical Analysis

The abundance of all the functional genes was calculated as previously described [18], namely using the transcripts per million kilobases (TPM) method using the following equation:(1)$$\mathrm{TPM}=\frac{rg\times rl\times 10^{6}}{flg\times T}$$
(2)$$T=\sum \frac{rg\times rl}{flg}$$
where *rg* is the number of reads that can be blasted onto gene sequences, *rl* is the read length, and *flg* is the gene length. Gene abundance was considered significantly different between two groups when |log_2_ (foldchange)| > 1 and *p* < 0.05 using the *t*-test. Multiple comparisons among groups were calculated using one-way ANOVA in R 4.0.0 [19]. Data were considered significantly different when *p* < 0.05. Graphs in this study were generated using ‘ggplot2’ package in R [20].

## 3. Results and Discussion

### 3.1. Concentrations of Heavy Metals in Soil

The background concentrations of heavy metals in soil and pig manure were determined. As presented in Table 1, both the concentrations in soil and pig manure were low according to literature [14], indicating the absence of prior heavy metal contamination. Cu and Zn levels measured at the start and end (60 d) of the experiment approximated the added concentrations plus the background levels (Figure 1A). These levels conformed to the Chinese Standard Soil Environmental Quality Risk Control Standard for Soil Contamination of Agricultural Land (GB 15618–2018) that limits Cu at 100 and Zn at 250 mg/kg in agricultural fields at pH > 7.5. Moreover, available Cu and Zn were also low and stable over the course of the experiments (Figure 1B).

### 3.2. Bacterial α Diversity in Soil

We analyzed the microbial populations in our experimental soils using ACE and Shannon indices to calculate the richness and diversity of the soil bacterial communities, respectively. The result showed that both ACE and Shannon indices were highest for HCuZn treatment and lowest for Control, while there was no significant difference between NCuZn and LCuZn (Table 2). This indicated that the manure application significantly increased the diversity of the bacterial community in HCuZn. The reason might be ascribed to the “Hormetic effect” of heavy metals on soil microbes, namely the effect of high-dosage inhibition and low-dosage stimulation. Cu and Zn both have been found to have this effect on soil bacteria, which is bacterial taxa-dependent [21,22]. In this study, we chose two common levels of Cu and Zn in pig manure as previously reported [14], rather than extremely high concentrations. Thus, the concentrations of Cu and Zn might not be high enough to decrease the bacterial diversity in the soil after manure application. It is important to note that the concentrations of Cu and Zn in HCuZn were still below the permissible limit standard for agricultural soil in China (GB 15618–2018). Furthermore, the significant increase in bacterial α diversity in HCuZn might be due to the change in bacterial community structure (see below). However, further studies are needed to elucidate the dose-effect of Cu and Zn upon a bacterial community in the soil to verify and explain this.

### 3.3. Bacterial Community Composition in Soils

Soil microorganisms were reported to be sensitive to elevations in heavy metal content in soil [23]. We found a significant change in the bacterial community structure in every treatment compared with Control (Figure 2). For example, the relative abundance of Actinobacteria at the phylum level substantially increased in the manured soil treatments compared with Control. In the group Control, the predominant taxa were Proteobacteria (53.34 ± 3.52%) and Actinobacteria (24.66 ± 0.90%), while in the manured soils, Actinobacteria were predominant (~60%) and Proteobacteria decreased to <40%. Moreover, the more metals were added, the more predominant the Actinobacteria tended to be (Figure 2A). The Actinobacteria possess rigid cell walls and can rapidly adapt to new environments, such as heavy metal pollution [24]. On the other hand, the extent of bacterial community structure alteration between treatments indicated that the primary change was caused by the original manure application, rather than heavy metal amendment (Figure 2A).

The most abundant bacterial genera that significantly increased following manure addition were *Streptomyces* (class: Actinobacteria), *Pseudomonas* (class: Gammaproteobacteria), *Glycomyces* (class: Actinobacteria), *Paracoccus* (class: Alphaproteobacteria), *Mycolicibacterium* (class: Actinobacteria), *Actinospica* (class: Actinobacteria), *Cellolosimicrobium* (class: Actinobacteria), *Luteimonas* (class: Gammaproteobacteria), *Micromonospora* (class: Actinobacteria), and *Rhodococcus* (class: Actinobacteria). Moreover, *Streptomyces* spp. in group HCuZn (21.40 ± 0.63%) significantly increased in relative abundance over that in NCuZn (19.74 ± 0.53%), indicating a positive effect of HCuZn treatment on this genus. This is consistent with the report that concluded *Streptomyces* spp. contained metal-resistance genes and could tolerate heavy metals in the environment [25,26] (Figure 2B).

There were genera whose relative abundance significantly declined after manure application. These genera included *Sinorhizobium* (class: Alphaproteobacteria), *Bacillus* (class: Bacilli), *Lysobacter* (class: Gammaproteobacteria), *Ensifer* (class: Alphaproteobacteria), *Arthrobacter* (class: Actinobacteria), *Nocardia* (class: Actinobacteria), *Variovorax* (class: Betaproteobacteria), and *Cupriavidus* (class: Betaproteobacteria). Although there are members of *Sinorhizobium* [27], *Bacillus* [28], *Lysobacter* [29], *Nocardia* [30], *Variovorax* [31], and *Cupriavidus* [32] that are resistant to heavy metals, our results indicated that these genera are not as tolerant to Cu and Zn as other bacteria.

### 3.4. Comparison of Differential Resistance Genes between Soil Groups

A total of 267 differential ARGs were identified between Control and NCuZn, among which 204 were significantly higher in abundance in NCuZn (Figure 3A,B). Similarly, there were 104 differential BMRGs between Control and NCuZn, among which 91 were significantly more abundant in NCuZn than Control. This indicated that, even in the absence of added metals and antibiotics, pig manure significantly influenced ARG and BMRG composition. This is consistent with the suggestion that the higher abundance of resistance genes in manured soil could be due to the manure application, even though the manure did not contain antimicrobials [33]. In contrast, comparisons between NCuZn and LCuZn, as well as between NCuZn and HCuZn, resulted in the identification of only 36 and 50 differential ARGs, respectively. Similarly, only five and six differential BMRGs were identified between NCuZn and LCuZn, as well as between NCuZn and HCuZn. These results indicated that, when considering the input of resistance genes from manure to the soil, it may primarily be ascribed to the animal manure itself, rather than to the heavy metals within the manure. The input of bacteria and ARGs along with ARG transfer and other environmental factors such as nutrient conditions were most likely the reasons that >200 differential ARGs were more abundant in NCuZn than in Control. When metal content increased in the manure, they further influenced the ARG profile but with less significance compared with the disparity between Control and NCuZn.

The transmission of resistance without antibiotic or metal pressure was expected because resistance genes are ancient, and most of the known ARGs were derived from the natural environment [34]. The existence of ARGs has even been frequently reported in pristine environments with no or little anthropogenic impact [35,36]. Moreover, ARG transfer is more likely to occur in areas where bacteria live at high density [37]. These findings help explain the reason why most differential genes (204/267 for ARGs and 91/104 for BMRGs) increased in the soil after manure application (Figure 3A,B). This implies an overall increase in the risk of resistance transmission in soil after manure application. In addition, there were 14/36 differential ARGs that increased in abundance in LCuZn between NCuZn and LCuZn. Similarly, there were 27/50 differential ARGs that increased in abundance in HCuZn between NCuZn and HCuZn. This suggested that the increase of heavy metals in manured soil might increase the expression of some ARGs and decrease the expression of some other ARGs. Based on these results, the common levels of heavy metals alone in pig manure from intensive farms might not cause an overall increased risk of ARG transmission in manured soil and perhaps, an overall decline if we use ARG abundance to assess this risk. However, since heavy metals cannot be degraded, their influence on ARG transmission in soil may be long-lasting, while a temporary effect of inhibition or promotion on ARG transmission may not stay put forever.

Differential ARG profiles of our experimental soils based on the class of resistant antibiotics indicated that multidrug-resistance was the most-frequent differential ARGs (Figure 4). This highlighted the risk of AMR transmission caused by pig manure application. Other important differential ARGs included those encoding resistance to aminoglycoside, cephalosporin, and fluoroquinolone. It is interesting to see that fluoroquinolone resistance so frequently differed between NCuZn and Control, because fluoroquinolone is a pure synthetic antibiotic, yet applying pig manure without antibiotics could induce so many of them into the soil. Most differential fluoroquinolone-resistant genes in our study were plasmid-encoded quinolone-resistance determinants (qnr genes). These genes have been found on plasmids and chromosomes and are presumed to be ancient in bacteria and may possess functions apart from quinolone/fluoroquinolone resistance [38].

### 3.5. Mobile Genetic Elements in Soil

MGEs, including plasmids, ISs, integrons, and ICEs, are responsible for the transfer of AMR genes within and between bacterial populations. In particular, plasmid conjugation is the predominant HGT mechanism, it plays a fundamental role in the dissemination of plasmid-borne ARGs [39,40]. ISs are short 1–2 kb MGEs with a simple genetic organization and are capable of inserting at multiple sites in a target DNA [41]. ISs contribute substantially to the generation of genetic diversity and can help promote the adaptation of the microbial hosts [42]. Integrons play a major role in the acquisition and dissemination of ARGs [43] and are found in ~6% of all sequenced bacterial genomes [44]. ICEs are widespread chromosome-borne MGEs in bacteria, which encompass all self-transmissible integrative and conjugative mobile elements, regardless of their mechanisms of integration or conjugation. ICEs have been found to bestow a wide range of phenotypes on the bacteria, including antibiotic and heavy metal resistance [45]. ICEs occupy up to 25% of bacterial genetic material and are the major promoters of genetic diversity in bacteria [46].

In our present study, 1274 differential plasmids were identified between Control and NCuZn, 1008 of which significantly increased in abundance in NCuZn compared with Control (Figure 3C,D). These levels were substantially higher than for all other types of MGEs, and they could be ranked as plasmids > ISs > integrons > ICEs. This order helps us understand which type of MGEs could be the most active following manure application and should gain our attention. Interestingly, the significant variation in ICE abundance was exceedingly rare after the heavy metals were spiked among manured soils. This may be associated with the passive propagation of ICEs via chromosomal replication, segregation, and cell division [47]. When ICEs transfer horizontally, they not only integrate site-specifically and mostly at conserved chromosomal target sites but also transfer at low frequencies [48]. Therefore, their HGT is limited compared with the chromosome-independent plasmid conjugation. In addition, these differential gene numbers again support the idea that the application of pig manure itself was the primary cause of MGE alterations, on which basis the addition of Cu and Zn would cause a much less variation of MGEs. 

These above results emphasized that the AMR transmission risk is primarily the result of manure application, and it was marginally altered by the presence of metal added in the pig industry. Given this, further studies are needed to explore the major driving factors for ARG transmission in the environment, such as bacterial community and nutrition conditions. Statistical methods, such as redundancy and variation partitioning analysis, have been applied, which indicated that the soil bacterial community rather than soil physiochemical properties was a more decisive factor influencing ARG transmission [49]. However, a recent study came to the opposite conclusion, that the bacterial community was not the decisive factor in the fate of ARGs [50]. In another study, only a weak correlation was found between the bacterial community and ARG profiles in a reservoir [51]. Indeed, heavy metals as a stressor can facilitate ARG horizontal transfer [52]. This potential risk is one of the major reasons that Europe has restricted Cu [12] and ZnO [13] addition to pig feeds in recent years. However, our present results suggest that the common levels of heavy metals in manure alter the resistance and MGE compositions to a certain extent but may not be the primary driving factors for ARG transmission. Although both Control and NCuZn contained background concentrations of Cu and Zn that could not be excluded in this study, their existence was natural, and their concentration was low. Therefore, we postulate that it is not necessary to pay much attention to the effect of background heavy metals on AMR transmission.

In addition, the heavy metal levels in pig manure we set were approximate to the average concentrations in different regions in the world [14]. However, much higher levels were found in different areas or pig feedlots. For example, an extremely high level of Cu 4223.75 mg/kg and Zn 10,603.23 mg/kg (dry weight) in pig manure was reported in an investigation carried out in Jiangxi Province, China [53]. The effect of such an extremely high level of heavy metals in pig manure on AMR in manured soil is currently unexplored. It is worthy of note that extremely high heavy metal concentrations will most likely cause a more significant influence on ARM transmission in manured soil. Furthermore, in reality, manure fertilizer is usually repeatedly applied to agricultural soils, and its long-term effect and linkage to AMR transmission should be elucidated.

Heavy metal speciation is also an important aspect for investigating AMR transmission, since heavy metals excreted by animals are not in the same form as those added to feeds [54]. The effect of different heavy metal speciation on AMR transmission from manure to soil is currently unknown. Thus, to further assess whether heavy metals influence AMR, whole-chain studies from feed to manured soil with low to high heavy metal levels would be extremely useful.

## 4. Conclusions

Pig manure application significantly shifted bacterial community structures; the Actinobacteria especially substantially increased in relative abundance. On this basis, the addition of common levels of Cu and Zn only marginally influenced the composition of the bacterial community, as well as ARGs, BMRGs and MGEs. This indicated that, while the heavy metals in manure could influence the transmission of ARGs in manured soil, our work once more emphasized the contribution to the transmission risk of resistance genes in soil made by organic fertilizer of pig manure. On the other hand, the levels of Cu and Zn we added were not extremely high levels in literature and did not constitute a high level of pollution. Therefore, future studies are necessary to assess the effects caused by a wider range of heavy metal levels on AMR transmission in the environment. Furthermore, more comprehensive and effective measures are needed to be taken to tackle the spread of ARGs and BMRGs in animal husbandry, since the benefits of the simple restriction of heavy metals in feeds will most likely not be satisfactory.

## Figures and Tables

**Figure 1 ijerph-19-11265-f001:**
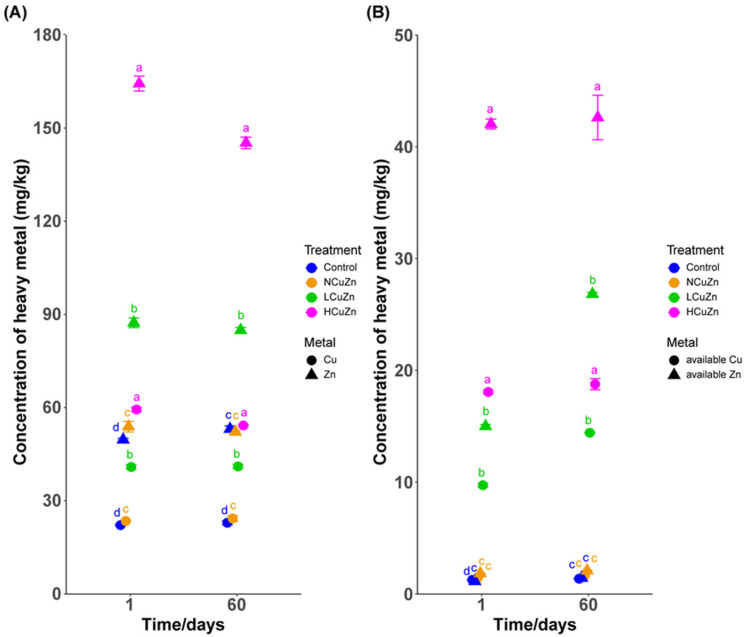
Heavy metal content in the experimental soils used in this study. (**A**) Total and (**B**) available Cu and Zn. Lowercase letters indicate statistical significance for treatment comparisons (ANOVA, *p* < 0.05).

**Figure 2 ijerph-19-11265-f002:**
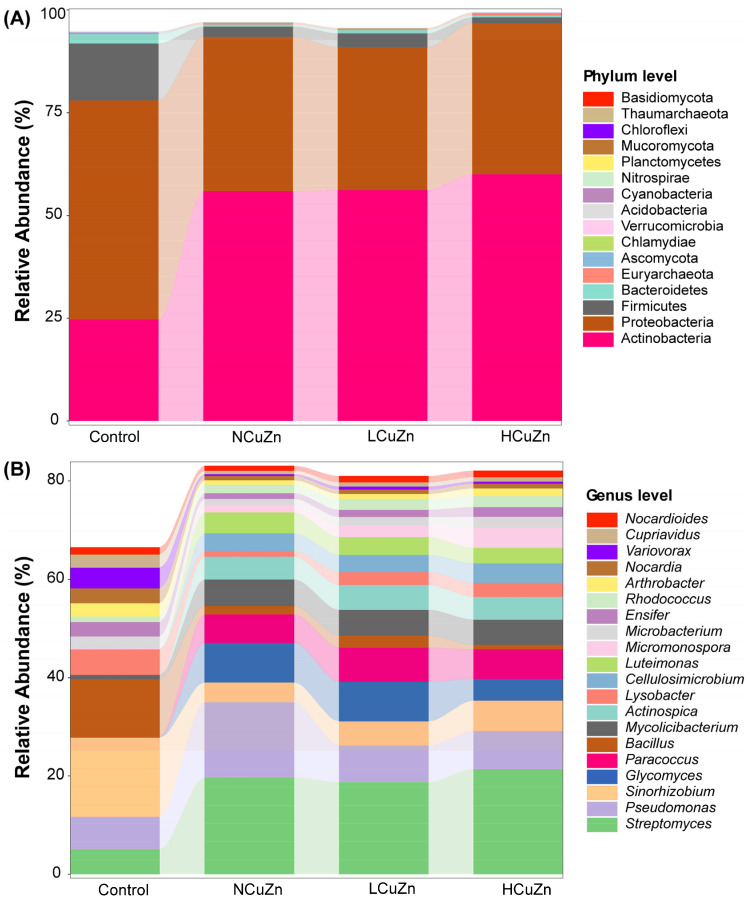
Bacterial community composition in different treatments. (**A**) Phylum level, (**B**) genus level.

**Figure 3 ijerph-19-11265-f003:**
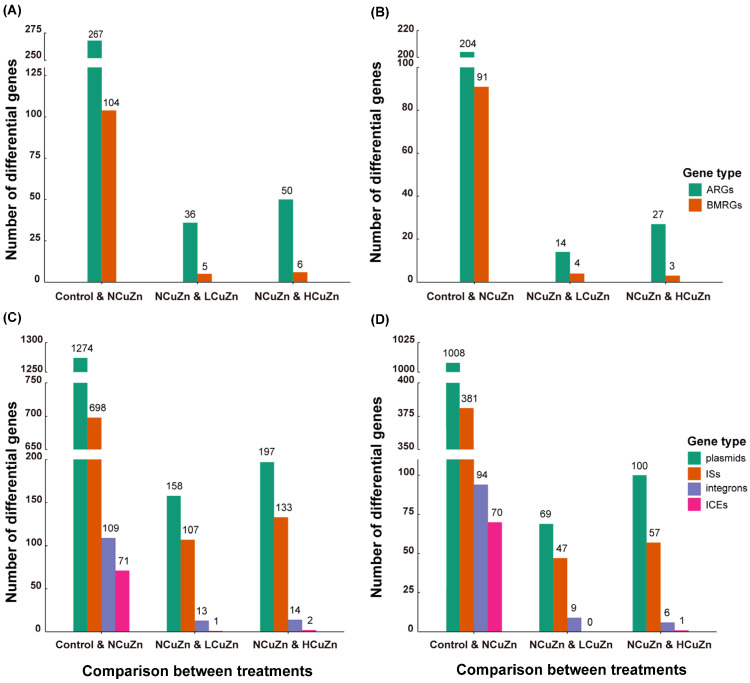
Number of differential genes between the experimental groups used in this study. (**A**) Total numbers of differential ARGs and BMRGs, (**B**) numbers of differential ARGs and BMRGs that increased in abundance due to manure application or heavy metal amendment, (**C**) total numbers of differential MGEs, and (**D**) numbers of differential MGEs that increased in abundance due to manure application or heavy metal amendment.

**Figure 4 ijerph-19-11265-f004:**
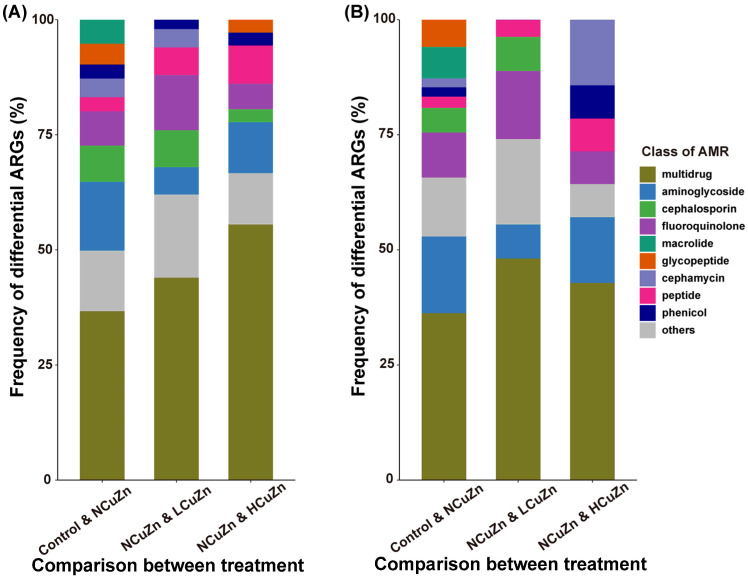
Differential ARG profiles between the experimental groups used in this study. (**A**) Frequency of total differential ARGs between treatments (%), (**B**) frequency of differential ARGs (%) that increased in abundance because of manure application or heavy metal amendment. AMR, antimicrobial resistance.

**Table 1 ijerph-19-11265-t001:** Background heavy metal content in original soil and pig manure (mg/kg dry weight).

Metal	Cu	Zn	Cd	As	Pb	Ni
Soil *	22.16 ± 0.29	49.60 ± 0.44	0.16 ± 0.02	ND	ND	ND
Pig manure	27.48 ± 1.29	93.14 ± 6.04	0.92 ± 0.04	0.87 ± 0.07	7.07 ± 0.35	7.44 ± 0.54

* Mean ± SD. ND, not determined.

**Table 2 ijerph-19-11265-t002:** Bacterial community richness and diversity indices in soil treatments.

Index	ACE	ACE Significance	Shannon	Shannon Significance
Control	202.33 ± 6.03 ^1^	C ^2^	5.54 ± 0.13	C ^2^
NCuZn	620.67 ± 7.02	b	6.04 ± 0.064	B
LCuZn	656.67 ± 44.46	b	6.21 ± 0.064	B
HCuZn	757.67 ± 35.53	a	6.47 ± 0.036	A

^1^ Mean ± SD. ^2^ Lower and upper letters indicate significance between groups (ANOVA, *p* < 0.05).

## Data Availability

Not applicable.

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
