# Peer review of "Controlling AMR in the Pig Industry: Is It Enough to Restrict Heavy Metals?"

_ijerph, 2022, doi:10.3390/ijerph191811265_

Round 1
Reviewer 1 Report
The article justifies pig manure a medium for heavy metals to transmit anti microbial resistance. This has been explained by an in-detailed work on metagenomics. However, there could be some minor revisions/modifications.
Introduction:
The authors could include a discussion on methods to tackle the anti-microbial resistance as a minor inclusion
Ghosh, D., Pramanik, A., Sikdar, N. et al. Synthesis of low molecular weight alginic acid nanoparticles through persulfate treatment as effective drug delivery system to manage drug resistant bacteria. Biotechnol Bioproc E 16, 383–392 (2011)
The ill effects of zinc as heavy metals and their presence in agricultural soils could be could be justified with appropriate references:
Basak, G., Das, D. & Das, N. Enhanced Zn(II) uptake using zinc imprinted form of novel nanobiosorbent and its application as an antimicrobial agent. Korean J. Chem. Eng. 31, 812–820 (2014).
Das, D., Vimala, R. & Das, N.Screening of Macrofungi for the Removal of Ag (I) and Zn (II) Ions from Aqueous Environment.Research Journal of Pharmaceutical, Biological and Chemical Sciences. 5(6),322(2014)
Materials and Methods:
Lines 70-72: Authors mention 'Top soil with no history of manure addition'. This statement could be a question since there could be traces of manure or fecal matter that the authors may not be aware of. I suggest to modify the statement.
Lines 82-86: The reasoning behind adding 4% manure is not clear. If the authors say that pig manure is contaminated with heavy metals, it could be better to work with samples and estimate the levels of copper and zinc rather than spiking the samples. If spiking is necessary, it could also be great to try on real contaminated soil samples. This could provide an insight on the realtime scenario
Results and Discussion
Table 1 shows the presence of As in soil samples and pig manure. the presence of 0.87 mg/kg of As could be a matter of concern as the acceptable limit is less than 5 ppb in water. The elemental form of As is not mentioned.
Figure 1: Representation of heavy metal concentrations is not clear. May be the graphs could be changed to another form like bar graph
Figure 2 has been represented well!
Reviewer 2 Report
1. Line 74 Please state the heavy metals concentrations and nutrients contents in the pig manure.
2. Line 83 How many treatments are carried out in this experiment? Treatments should be conducted in triplicate.
3. Line 90 The incubated period is too short to reflect well the accumulation of heavy metals and the effects of bacterial diversity.
4. Line 94-97 Please change “HNO3, HClO4 and CaCl2” to “HNO3, HClO4 and CaCl2”.
5. Results and discussions 3.1. Concentrations of heavy metals in soil
Line 154-156 “Background heavy metal content in original soil and pig manure” may be presented in material and methods.
6. Line 180 Table 2 Please indicate the meaning of upper and lower case letters
7. Standard errors of means should be present in Figure 3.
